

# Does the AO index have predictive power regarding extreme cold temperatures in Europe?

Tamás Bódai[1,2] and Torben Schmith[3]

[1]Center for Climate Physics, Institute for Basic Science, Busan, Republic of Korea, 46241
[2]Pusan National University, Busan, Republic of Korea, 46241
[3]Danish Meteorological Institute, Copenhagen, Denmark

**Correspondence:** T. Bódai (bodai@pusan.ac.kr)

**Abstract.** With a view to seasonal forecasting of extreme value statistics, we apply the method of Nonstationary extreme value statistics to determine the predictive power of large scale quantities. Regarding winter cold extremes over Europe we find that the monthly mean daily minimum local temperature – which we call a native co-variate in the present context – has a much larger predictive power than the nonlocal monthly mean Arctic Oscillation index. Our results also prompt that the exploitation
of both co-variates is not possible from 70 years long data sets.

## 1   Introduction

A new capability of extending the lead time for weather forecast will always find users. This quest has a fundamental barrier, however: chaos (Tél and Gruiz, 2006). The nonlinearity and complexity of fluid dynamics make the system evolution sensitive
to initial conditions. But in weather forecast the issue is not only that measurements are associated with a finite "accuracy"; the continuum of the field variables are sampled discretely, and very sparsely too above the oceans and in the vertical dimension, and ignoring completely some physical quantities. Therefore, with new capabilities of closing the gap on sparsity and with attention to more quantities, thanks e.g. to new weather satellites, we can still make gains for forecast skills. Another avenue of gaining skill is via probabilistic forecasting (Dobrynin et al., 2018). This requires computing power, as an ensemble of
possible evolutions – the more the better – needs to be followed. Furthermore, the skill depends also on the "scale of the observable" (Gálfi et al., 2017); giving up on more detailed information, coarser quantities of spatial or temporal averages can be more skillfully predicted. Seasonal forecasting is emerging as probabilistic and targeting coarse properties (MacLachlan et al., 2015; Scaife et al., 2014).

Instead of temporal averages, we are interested in the predictability of extreme tail probabilities, i.e., the probabilities of
high/low threshold exceedances. Like the Central Limit Theorem concerning averages, maximal/minimal elements in batches or threshold exceedances also obey limit laws, namely, the Generalised Extreme Value (GEV) distribution and the Generalised Pareto (GP) distribution, respectively (Fisher and Tippett, 1928; Coles, 2001). This is very favourable considering that a finite



ensemble could be insufficient to characterise a probability distribution with respect to its *tail* in particular. Instead, Extreme Value Theory is providing us parametric models of these tails.

The climatological GEV distribution

$$\mathrm{GEV}(z; \mu, \sigma, \xi) = \exp[-[1 + \xi(\frac{z - \mu}{\sigma})]^{-\frac{1}{\xi}}],$$

$\mu = \mu_s$, $\sigma = \sigma_s$, $\xi = \xi_s$ ('s' for 'stationary'), can be inferred from long-term data, for any particular location and observable. However, it would be desirable to forecast the parameter values that apply more specifically to, say, the forthcoming winter. That is, we consider the monthly extremum as a nonstationary random variable:

$$Z_t \sim \mathrm{GEV}(\mu(t), \sigma(t), \xi(t)).$$

The time-dependence can be thought of as one that takes place via a time-dependent so-called "co-variate" (Coles, 2001). In an analogous situation Friederichs and Thorarinsdóttir (2012) applied the same approach to forecast daily peak winds. Adopting the idea of *sharpness* of a probabilistic forecast (Gneiting et al., 2007), we regard the co-variate to have *predictive power* if the expected "forecast" *scale* parameter is smaller than the climatological one: $\sigma_s > \langle \sigma(t) \rangle$. To put this particular concept of predictive power into perspective, we note that in practice one would seek a co-variate with high predictive power, and, the

seasonal forecast would be performed, with, say, a month or so lead time, to predict the co-variate as a large scale quantity. This way the skill of forecasting the quantity of interest would derive from 1. the skill of forecasting the co-variate, 2. the (theoretical) predictive power of it, and, furthermore, 3. the size and quality of the data set from which the model of co-variate-dependence is inferred and 4. the approximation of the truth by that model [1] (see Sec. 2). Here, however, we will focus on the predictive power of co-variates only; that is, we do not assess, say, the forecast skill of any seasonal forecast system.

As a case study, we consider extreme cold winter temperatures in Europe. Our analysis of the predictive power of co-variates is based only on *observational* (or more precisely, see below, gridded/E-OBS) data, that is, no circulation model simulation output data is considered in order to infer the prediction model. It has two consequences we wish to mention here. Firstly, we can make conclusions about the real weather/climate system. Secondly, with a view to "predictability", the potentially coarse spatial resolution of a seasonal forecast system would be rendered irrelevant, because it is used only to forecast the large scale

co-variate, while the co-variate-dependence is inferred from the observational data that is native in every sense – including the spatial scale – to the quantity to be forecast. The analysis is carried out for individual locations represented by gridpoints, and an extensive part of Europe as well as North Africa is covered, drawing on the European Climate Assessment & Dataset (ECAD, www.ecad.eu) project (Klein Tank et al., 2002; Cornes et al., 2018). We find that the Arctic Oscillation (AO) index as a co-variate has some predictive power regarding extreme cold temperatures in Europe. However, our main finding is that the

monthly average daily minimum temperature (versus the AO index) has a lot more predictive power.

Cold winter temperature extremes in Europe (Sillmann et al., 2011) and North America (Whan et al., 2016) have been studied using the same methodology but with a blocking indicator as a co-variate. These studies compared extremes in climate

---

[1] The authors Friederichs and Thorarinsdóttir (2012) call this model a "prediction model", although they acknowledge that in itself it establishes nowcasting only. Furthermore, we use the term "predictive power" in the same sense as theories make predictions about what-if scenarios. We also note that strictly we should speak about the predictive power of a "prediction model" instead of just that of a co-variate.





models and reanalysis data with an interest in model fidelity. Similarly, Whan and Zwiers (2017) studied the ability of regional climate models to reproduce the (tele)connection of extreme winter precipitation in North America with the North Atlantic Oscillation (NAO) and El Niño–Southern Oscillation (ENSO) phenomena.

The rest of the paper is organised as follows. In Sec. 2 we outline our methodology based on Nonstationary extreme value statistics, proposing four nonstationary models of GEV parameter-co-variate-dependence. In Sec. 3 we present results encompassing Europe, comparing the performance of the four models both with respect to predictive power and goodness-of-fit. In Sec. 4 we discuss our results and provide an outlook for future work.

## 2 Methodology

### 2.1 Nonstationary extreme value statistics

Coles (2001) observed that the annual maximum sea-levels at Fremantle are greater in years when the average Southern Oscillation index (SOI) is higher, and performed a so-called *nonstationary* extreme value statistics (EVS) using the SOI as a co-variate. The latter simply involved a "model" being a functional relationship of some parameter of the GEV distribution and the co-variate. This model features parameters to be inferred from data – time series of the observable of interest, on the one hand, and time series of the co-variate, on the other. For the inference, the Maximum Likelihood Method (MLE) can be applied.

The influence of the Arctic Oscillation on regional weather has been studied e.g. by Wettstein and Mearns (2002); Hu and Feng (2010). Recently, following (Bódai et al., 2020), this influence has been reevaluated by Haszpra et al. (2020) via mapping out correlations of the AO index with local temperature and precipitation, using a conceptually sound ensemble-based methodology (Drótos et al., 2015). In particular, seasonal mean temperatures in Northern Europe correlate with the seasonal mean AO index. This is also indicated by the diagram in Fig. 1 (a) showing the daily minimum temperature in Kiev vs the monthly mean AO index ($AO$) in a scatter plot. We might say that the average temperature depends on negative but not really positive values of $AO$. Our novel observation is that for extremes this is the other way round: they depend also on positive $AO$ values. Furthermore, the overall dependence of e.g. the location parameter $\mu(AO)$ of a "nonstationary GEV distribution" is seemingly *nonmonotonic*. Therefore, as a minimal model we propose to use a model with quadratic parameter-dependences:

$$\xi(t) = \xi_0, \tag{1}$$

$$\mu(t) = \mu_0 + \mu_{1,AO}(AO(t) - AO_0) + \mu_{2,AO}(AO(t) - AO_0)^2, \tag{2}$$

$$\sigma(t) = \sigma_0 + \sigma_{1,AO}(AO(t) - AO_0) + \sigma_{2,AO}(AO(t) - AO_0)^2. \tag{3}$$

Note that we model the shape parameter as a constant, as its inference is known to be more sensitive to data scarcity (Friederichs and Thorarinsdóttir, 2012), but also theory (Holland et al., 2012; Lucarini et al., 2014, 2016; Bódai, 2017) dictates it, at least under stationary climate, as we explain shortly. We will refer to this model as Model #1 or M1. As shown by Fig. 2 (b), one month looks to be a long enough period (or "block") to yield temperature minima that already conform well to the GEV distribution.




We implemented our code for Nonstationary EVS in Matlab using `mle`. This code was applied recently to an analogous problem for estimating the potential barrier height from escape time data (Bódai, 2018). Like Matlab's `gevfit`, we perform the rootfinding for $\ln\sigma$ in order to prevent the rootfinder algorithm to select wrong negative values for $\sigma$.

We note that the model M1 is almost certainly wrong (apart from the problem of the finite block size of a month used), even if providing a good approximation. We can explain this by considering two aspects of the situation, the second of which

to be identified as the one associated with the approximation. Firstly, looking at the dependence of a parameter of the GEV distribution on a co-variate assumes that for any possible fixed value of the co-variate the variable of interest is really distributed according to a GEV distribution. This is actually correct, because 1) the co-variate of any predictive power is some function of the state variables of the considered dynamical system, and 2) sampling the evolution according to some fixed value of it amounts to introducing a Poincaré surface of intersection, yielding a new discrete dynamical system. Actually, already

the block maxima of a time-continuous dynamical system are associated with a Poincaré section (Bódai, 2015, 2017). 3) A recent theory (Holland et al., 2012; Lucarini et al., 2014, 2016) establishes that observables of dynamical systems feature an extreme value law – or something resembling that with a very good approximation for high-dimensional systems (Bódai, 2017). Interestingly, different Poincaré sectioning surfaces associated with different fixed values of the co-variate should yield sections of the attractor whose geometries are different, yet, their fractal dimensions (Hall and Davies, 1995) and so the

shape parameter (Holland et al., 2012; Lucarini et al., 2014; Bódai, 2017) should be the same. Nevertheless, considerable variation of finite-size estimates are possible, even if the pre-asymptotic non-GEV characteristics does not show up (Bódai, 2017; Gálfi et al., 2017). Secondly, unlike the shape parameter, the location and scale parameters should depend in a particular way on the co-variate, which is almost certainly not expressible by elementary functions like it is practical to assume for inference purposes. Yet, some models are better than others, such that they might not even be rejected by suitable statistical

tests (Sec. 2.2).

We rely on further assumptions. There are two sources of nonstationarity that our methodology does not (and probably cannot) take into account. (We note that the term 'nonstationary process' is common to apply to stochastic processes, while deterministic dynamical systems are said to be 'nonautonomous'. Here we adhere to the choice of classical Extreme Value Theory.)

– Seasonality: the different months of the winter should have different climatologies (Bódai and Tél, 2012).

– Climate change: just like the seasonal cycle, 20th century human activity imposes an external forcing and is known to cause climate change, i.e., probability distributions and so expectations, etc., do change (Drótos et al., 2015).

Our assumption regarding both types of nonstationarity is that while the unconditional temperature distribution is more significantly affected, the probabilities *conditioned* on the co-variates are more robust in the considered situation. Furthermore,

we believe that even if even the conditional probabilities were somewhat affected, our conclusion on the predictive power of the co-variates as we define it here is rather robust to this. Otherwise, the effect due to seasonality would be a *conservative*





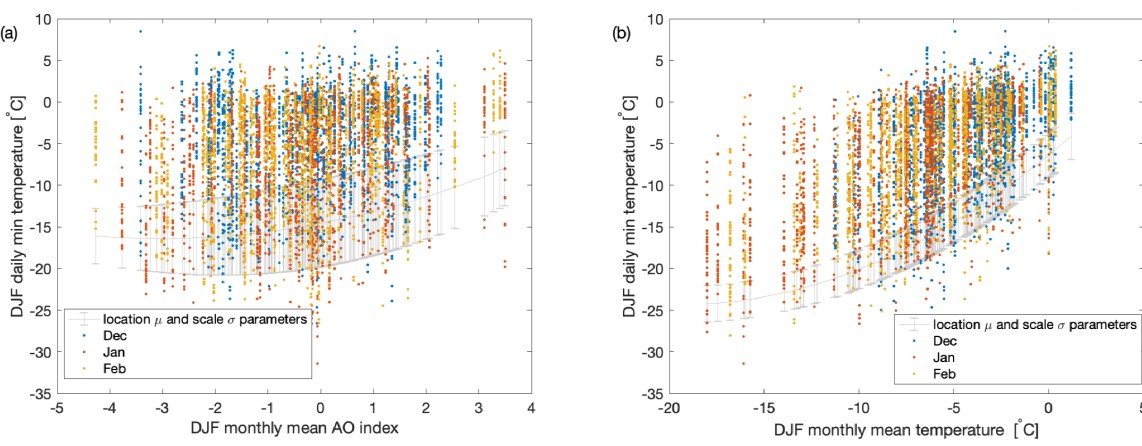

**Figure 1.** Scatterplots of daily minimum temperature at Kiev vs a co-variate. The co-variates are (a) the monthly mean AO index and (b) the monthly mean daily minimum temperature. We chose time series data from the ECAD gridded dataset at (latitude,longitude) = (50.25°N,30.25°E), the gridpoint nearest to Kiev. The inferred parameter-dependence given by Eqs. (2,3) are shown by a solid line $(\mu(AO))$ with "brackets" around it $(\sigma(AO))$ in the range of data availability.

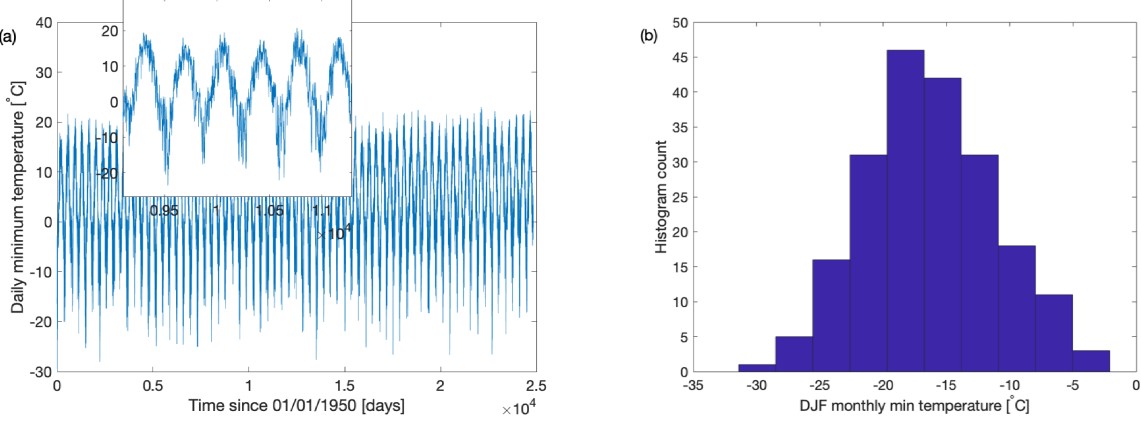

**Figure 2.** Temperature data for Kiev: (a) time series of daily minima, (b) distribution of monthly minima.





estimate of the predictive power, as, by pooling monthly data of different climatologies, i.e., data of a larger spread, we would have a larger scale parameter estimate than the true scale parameter belonging to any single winter month [2].

Furthermore, beside the climate-change-type nonstationarity, playing out on multidecadal time scales, there should be considerable internal variability on multidecadal time scales, too. This means that the $\sim 70$ years worth of ECAD data could not sample the climatology even if it was stationary. That is, histograms based on 70 years of data can be significantly different from the true climatological distribution. Regarding this problem stemming from internal variability, we have the same assumption as regarding nonstationarity: probabilities conditioned on the co-variate are much less affected by multidecadal internal variability.

It stands to reason that we have a better co-variate than the monthly mean AO index in the monthly mean temperature, or the monthly mean daily minimum temperature, at the same location, as it is 1) the same physical quantity and 2) pertains to the same location. We can call it a "native" co-variate. We rerun the inference using the same model as (1)-(3) but with the monthly mean daily minimum temperature $T$ as the co-variate, to be referred to as M2, and present the resounding results in Fig. 1 (b) on top of the scatter plot. While using the $AO$ there is just a small gain possible: $\sigma_s = 4.63 > \langle \sigma(t) \rangle = 4.41$, using $T$ we can, potentially, gain much more: $\sigma_s = 4.63 > \langle \sigma(t) \rangle = 2.39$.

As we found – at least for Kiev – that both $AO$ and $T$ have predictive power, we will also examine a quadratic model, M3, featuring both as co-variates:

$$\xi(t) = \xi_0, \tag{4}$$

$$\mu(t) = \mu_0 + \mu_{1,AO} AO^*(t) + \mu_{1,T} T^*(t) +$$
$$\mu_{2,AOT} AO^*(t) T^*(t) + \mu_{2,AO} AO^*(t)^2 + \mu_{2,T} T^*(t)^2, \tag{5}$$

$$\sigma(t) = \sigma_0 + \sigma_{1,AO} AO^*(t) + \sigma_{1,T} T^*(t) +$$
$$\sigma_{2,AOT} AO^*(t) T^*(t) + \sigma_{2,AO} AO^*(t)^2 + \sigma_{2,T} T^*(t)^2, \tag{6}$$

where e.g. $AO^*(t) = AO(t) - AO_0$. However, with M3 over M2 we make no gain on this occasion, as $\langle \sigma(t) \rangle = 2.68$; what is more, we have something like the opposite of a synergistic effect: the deterioration of the performance when trying to exploit co-variates that are "skilled" on their own. No doubt this is an overfitting problem, having too many parameters of the model to infer from too little data. To alleviate this problem, we attempt to remove some parameters. Experimenting with different possibilities (Matlab's `mle` did not return confidence intervals when including the shift parameters e.g. $AO_0$), we found the best result $\langle \sigma(t) \rangle = 2.29$ with enforcing $\sigma_{2,AO} = 0$, $\sigma_{2,T} = 0$ (M4). We should have a better idea of the systematic gains of M4 over M3 when we repeat this exercise for other gridpoints.

To do this we downloaded E-OBS gridded data from www.ecad.eu/download/ensembles/download.php, namely, daily minimum temperature (variable TN) time series spanning from 1950 to 2018. The lateral resolution of the regular grid is $0.5°$. The time series derive from station data measuring temperature at 2m height, by applying some interpolation technique (Klein Tank

---

[2]Note that taking monthly minima of a *nonautonomous* system is itself just an operative approach. Extreme value statistics for nonautonomous systems should be defined by the instantaneous snapshot attractor (Bódai and Tél, 2012), such that blocks are not temporal windows but the number of random samples from the ensemble that represents the snapshot attractor.




et al., 2002). However, in our analysis, like for the above example of Kiev, we treat individual time series separately. As for the co-variate, we downloaded the AO index from www.cpc.ncep.noaa.gov/products/precip/CWlink/daily_ao_index/ao_index.

html.

## 2.2    Model diagnostics

Given that we know that our models are strictly speaking wrong, as we explained in the previous subsection, we would like to check if they are useful at all. We will attempt to reject the hypothesis that the observational data is drawn from (something like) the distribution obtained by inference. We will use a common strategy (Coles, 2001) to possibly accommodate the non-

stationarity of the GEV distribution, $Z_t \sim \text{GEV}(\mu(t), \sigma(t), \xi)$: we "standardise" the random variables, for each observational data point $-z_i$ (the negative sign is applied to be able to deal with maxima instead of minima), by transforming them such that all of them are supposed to conform to the *same* GEV distribution with *constant* parameters, $\mu = 0$, $\sigma = 1$, for simplicity. We apply two different tests.

-   Chi-squared test: Does the transformed data $y_i$ obtained by the transformation

$$Y_t = (Z_t - \mu(t))/\sigma(t)$$

    conform to the "standardised" version of the inferred GEV distribution specifically (featuring the original $\xi$)?

-   Lilliefors test: Does the transformed data $x_i$ obtained by the transformation

$$X_t = \log(1 + \xi Y_t)/\xi$$

conform to any Gumbel ($\xi = 0$) distribution?

We use Matlab's `chi2gof` and `lillietest` to carry out these tests.

## 3    Results

As we regard the co-variate to have predictive power when $\sigma_s > \langle \sigma(t) \rangle$, first we map out the climatological EVS. The result of this can be seen in Fig. 3. The "variability" of extremes are characterized by the shape $\xi$ and scale $\sigma$ parameters: variability is

larger either when (a negative) $\xi$ is smaller (in modulus) or when $\sigma$ is larger. Interestingly, in the part of Europe shown, these characteristics of variability have opposite tendencies: e.g. $\xi$ tends to be greater where $\sigma$ is greater, and vice versa. The location parameter $\mu$ characterises the "disposition" for extremes. In Europe northward and deeper in the continent, in the Northeast, does the winter tend to be more inhospitable.

Next we present and compare the predictive powers of the models M1-M4 in *absolute* terms. We can see $\sigma_s - \langle \sigma_j \rangle$ (the larger

the better), $j = 1, 2, 3, 4$, mapped out in Fig. 4. It is clear that $AO$ as a co-variate has less predictive power than $T$ everywhere, much less in most places, and it does its best around the Baltic Sea. M2 systematically outperforms M3, and M4 makes hardly any appreciable gain. $T$ seems to perform best where $\sigma_s$ (Fig. 3(c)) is larger. Therefore, in Fig. 5 we plot, at least for M1,

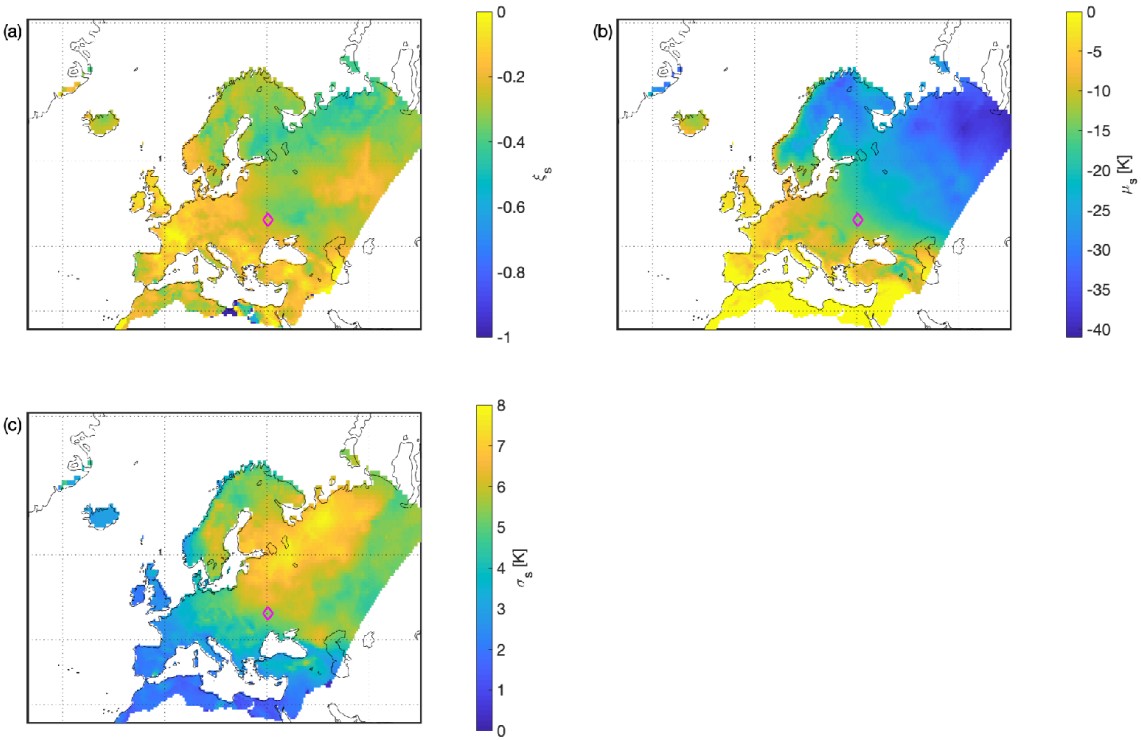

**Figure 3.** Climatological extreme value statistics based on the ECAD data, given by the (a) shape, (b) location, (c) scale parameters of the GEV distribution. A diamond in magenta marks the gridpoint located nearest to Kiev (see Figs. 1 and 2).

M2, the predictive power also in *relative* terms: $1 - \sigma_j/\sigma_s$ (the larger the better). The map for $T$ in panel (b) shows a fairly even relative power, suggesting that the "nativity" of $T$ as a co-variate dominates. Accordingly, the patterns of the absolute and relative powers of $AO$ are very similar.

Going beyond simply comparing maps, in Fig. 6 we (a) quantify the predictive power of $T$ relative to $AO$, i.e., M2 relative to M1, and also plot the differences showing the "gains" of trying to exploit both co-variates, (b) M3 and (c) M4, over using just the temperature, M2. We can see that M2 can outperform also M4, while any gain made by M4 over M2 is very little in quantitative terms. M3, on the other hand, can be significantly outperformed by M2. Most importantly, $T$ outperforms $AO$ many times over in relative terms, the more the further away from Northern Europe in the proximity of the North Sea and Baltic Sea.

Finally, we present the results of the chi-squared and Lilliefors statistical tests. The so-called p-values of these tests are shown in panels (a) and (b) of Figs. 7 and 8 for M1 and M2, respectively. Low p-values for the two tests, especially those less


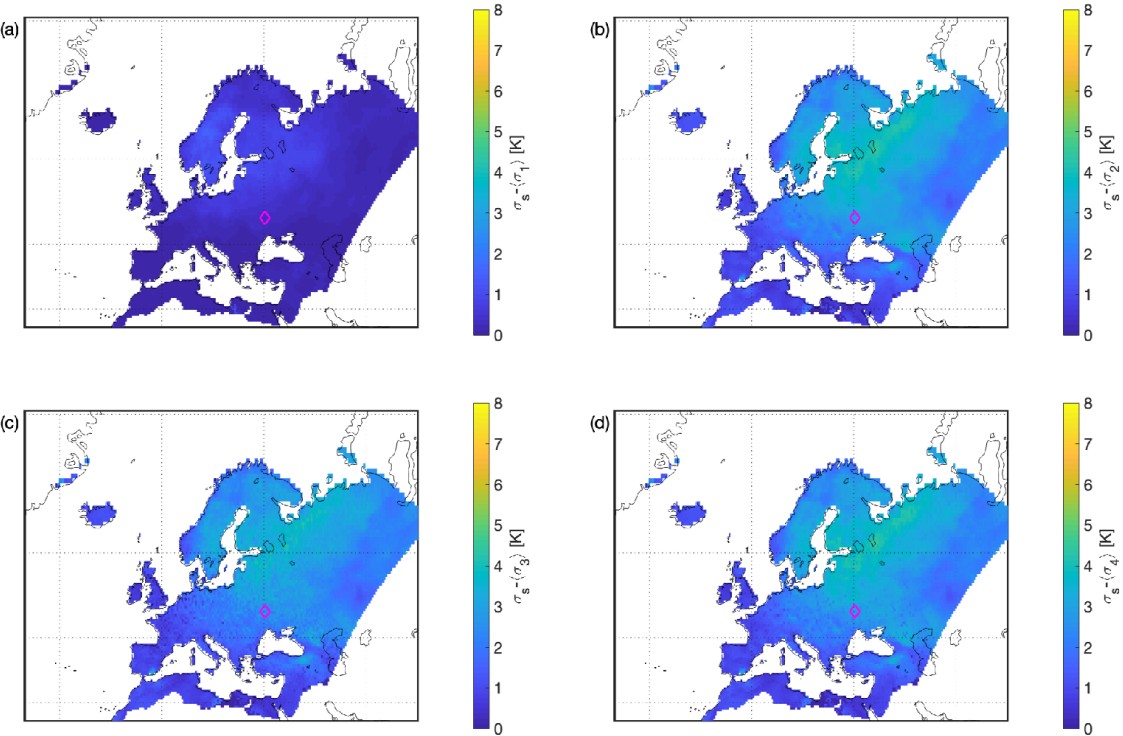

**Figure 4.** Predictive power of co-variates or models involving them regarding extreme cold temperatures in Europe, in absolute terms. The models are: (a) M1, (b) M2, (c) M3, (d) M4. For comparison, we kept the colour-value range of the climatological $\sigma_s$ plotted in Fig. 3 (c). All the maps are tessellated by monochrome tiles in respect of the gridpoint-wise analysis.

than 0.05 which is commonly taken to reject the null-hypothesis, tend to coincide. However, $\sim$80 % of the time either of the models cannot be falsified. When they can be, one might think that it goes together with a positive shape parameter $\xi > 0$, because the asymptotic value should be negative, and this nonasymptotic characteristics might show up also in a non-GEV shape of the distribution. However, to the contrary, while we find positive estimates for $\xi$, we do only for M2, and even these occurrences do not typically coincide with $p < 0.05$.

## 4 Discussion and Outlook

We applied Nonstationary extreme value statistics in the new context of determining the predictive power of co-variates regarding extreme events. In particular, we looked at the predictability of winter season cold extremes in Europe. This analysis


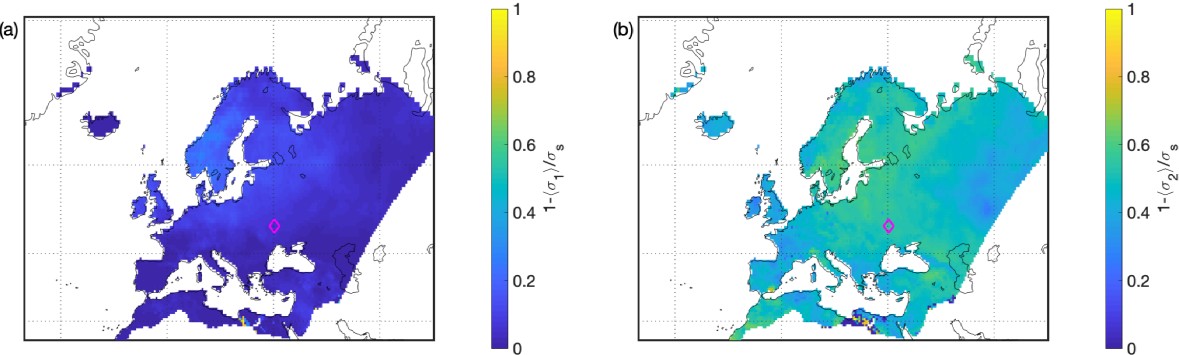

**Figure 5.** Predictive power of co-variates, monthly mean (a) AO index and (b) daily minimum local temperature, regarding extreme cold temperatures in Europe, in relative terms.

is based on daily "observational" (gridded E-OBS) data and Extreme Value Theory only; no seasonal forecast systems are evaluated. Our main conclusions – an encouraging one, on the one hand, and a negative result, on the other – are the following:

- A native co-variate will perform far better: the predictive power of the local monthly mean daily minimum temperature is many times better in Europe than the nonlocal monthly mean AO index.

- Trying to exploit both co-variates leads to deteriorating predictability as opposed to a synergistic effect, likely due to overfitting.

Even fairly unsophisticated quadratic models turn out to feature too many parameters to be inferred from a fairly short observational record. This data scarcity should be somewhat alleviated by "pooling" correlated data from neighbouring gridpoints, assuming a smoothness of the spatial dependence of the parameters of the models. So-called Max Stable processes are commonly used to model spatial extremes, see e.g. (Padoan et al., 2010; Ribatet, 2017), which model can be extended to feature nonstationarity (Huser and Genton, 2016). We will explore this avenue in future work.

Another way of dealing with data scarcity is to consider more coarse properties – with respect to either the probability density or its co-variate-dependence (or both). An analysis by van den Besselaar et al. (2009) of the ECAD dataset used more basic *indices* of extremes, and, instead of assuming continuous co-variate dependence, they studied a variation with respect to (discrete) so-called circulation types as well as "climate change" in terms of long-term trends. In a similar vein, Higgins et al. (2002) analysed some coarse relationship of cold and warm extremes over the US with the leading modes of climatic variability: the AO and ENSO.


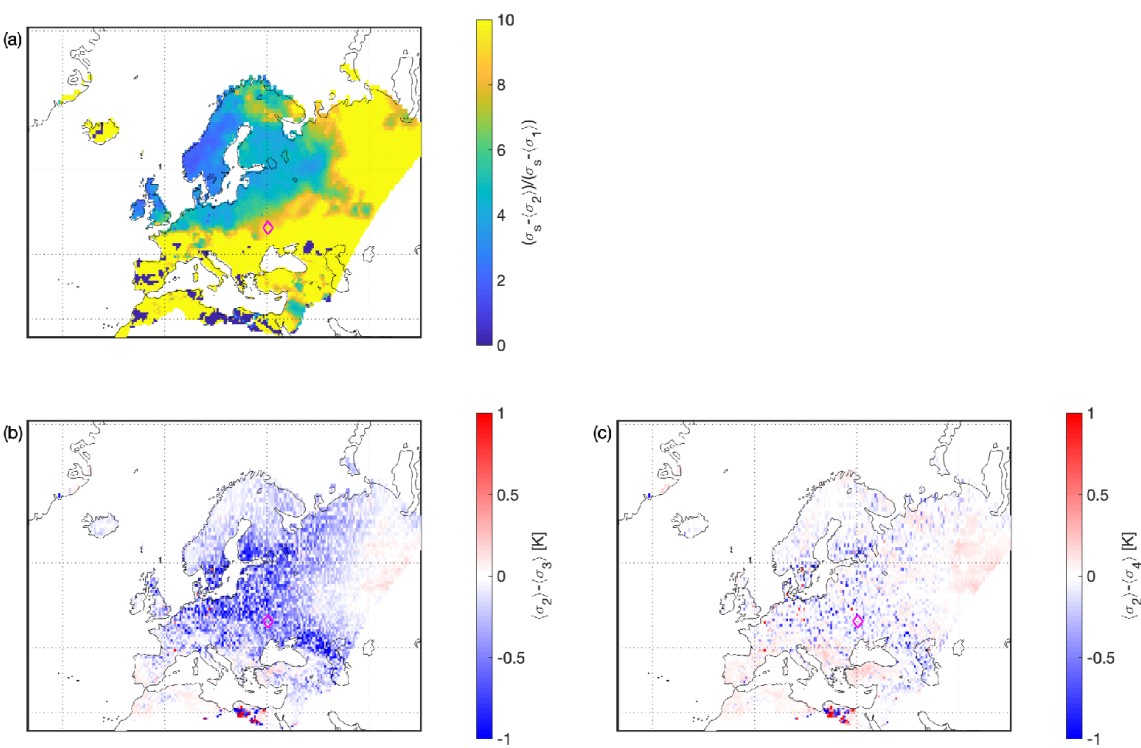

**Figure 6.** Comparison of the predictive powers of different models M1-M4 regarding extreme cold temperatures in Europe: (a) that of M2 relative to that of M1, and the absolute "gains" of (b) M3 and (c) M4 over M2. Cutoffs at the upper and lower ends of the colour range were applied.

In some gridpoints and for some model we were able to reject the GEV model. Therefore, we are prompted to check in the future if monthly temperature minima in circulation models, for which abundant data can be generated, also defy the GEV distribution.

In our analysis we relied on the assumption that probabilities conditioned on the co-variates considered are not affected much either by seasonal and long-term nonstationarity as a forced response, or by significant internal variability on multidecadal time scales. We also relied on theory that under a stationary climate the true shape parameter is constant. Nevertheless, a nonparametric statistical test has been recently devised to test the hypothesis of the constancy of the shape parameter (de Haan and Zhou, 2017). Clearly, it is meaningless to assume some dependence and carry out inference if the hypothesis of constancy cannot be rejected. It is also known that the estimation of the shape parameter of a stationary process is rather sensitive to data shortage (Friederichs and Thorarinsdóttir, 2012).


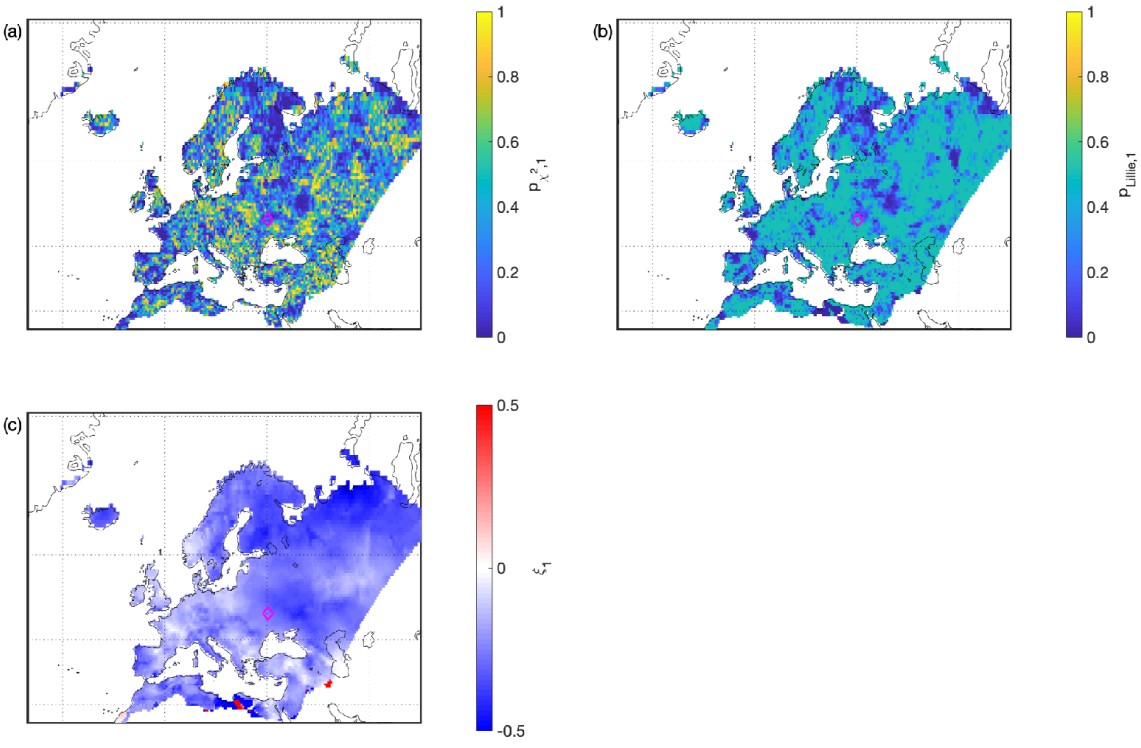

**Figure 7.** Evaluation of model M1. p-values of the (a) chi-squared and (b) Lilliefors tests, and (c) shape parameters.

It would be clearly desirable to exploit the finding that native co-variates have larger predictive power. In our case this
would mean that some seasonal forecast system can actually forecast the monthly mean daily minimum temperature, say, just
as well as the monthly mean AO. However, this might not be the case when going about forecasting naively. Nevertheless,
it has been demonstrated recently (Dobrynin et al., 2018) that seasonal forecast ensembles sampled according to how well
individual ensemble members predict the NAO index (assessed by a simple statistical model featuring four "predictors": SST
in the North Atlantic, sea ice volume in the Arctic, snow depth in Eurasia, stratospheric temperature at 100 hPa) displays a
significant improvement of its skill to predict 2m surface temperature in extensive parts of Europe. Given that the NAO (Hurrell
et al., 2013) is part of the AO phenomenon, and we found that the latter has much less predictive power than some average
temperature regarding cold extremes (Sec. 3), the said finding of Dobrynin et al. (2018) is a surprising and ironic one. However,
it is probably the circulation model to be credited for, bridging the gap between the two co-variates considered. Perhaps the
concept of the "predictive power" of a quantity can be defined less restrictively than how we did it for the present study, allowing
possibly for mixing it with the concept of "predictability". This way the answer to the question posed by the title of our paper


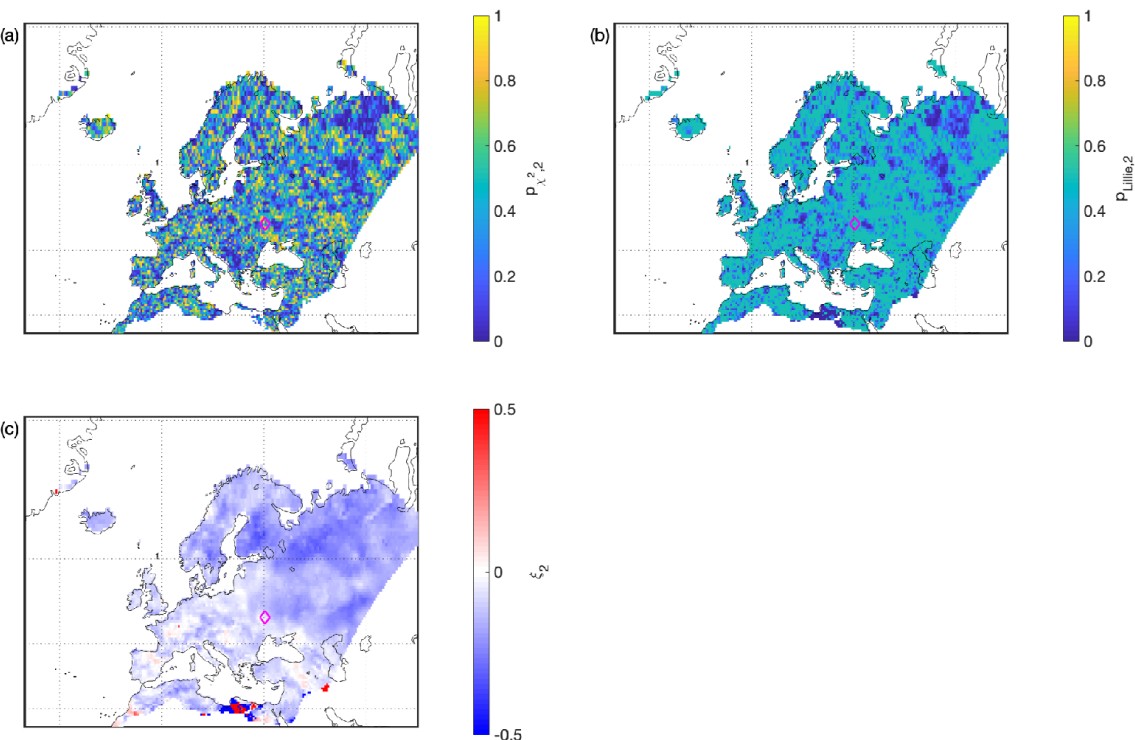

**Figure 8.** The same as Fig. 7 but for M2.

might be rather different. As future work, we intend to evaluate the skill of various seasonal forecast systems of predicting extreme value statistics using our scheme combined with the mentioned "NAO-targeted ensemble subsampling". Although the AO index is found to have less predictive power, if its seasonal forecast is much more skillful (Athanasiadis et al., 2017; Hall et al., 2017; Smith et al., 2016; Dunstone et al., 2016) than that of some average temperature, then it might be a competitive

alternative to the above suggestion. We intend to make this comparison. We suggests that our models of Nonstationary EVS can also be based on reanalysis data, whereby the EVS can be predicted also over seas, which could greatly benefit seafaring.

*Code availability.* The analysis codes are available from the corresponding author upon request.

*Data availability.* Data used for this analysis is available from the sources given in the main text (last paragraph of Sec. 2.1).





*Author contributions.* TB conceived the idea of applying Nonstationary EVS using native co-variates to the forecasting of EVS, and TS
suggested to make a comparison with using the non-native co-variate of the AO index. TB performed all the analysis and created all the
figures. Both authors contributed to the preparation of the manuscript.

*Competing interests.* The authors declare no conflict of interest with any parties.

*Acknowledgements.* We acknowledge useful interactions with Magdalena Alonso Balmaseda, Stefan Siegert, and colleagues on the Blue-
Action project: Shuting Yang, Martin King, Panos Athanasiadis, and thank Johanna Baehr and Øivin Aarnes in addition for their feedback on
the first version of the manuscript. We acknowledge the E-OBS dataset from the EU-FP6 project UERRA (www.uerra.eu) and the Copernicus
Climate Change Service, and the data providers in the ECA&D project (www.ecad.eu). This work received financial support from the Blue-
Action project (under grant No. 727852) and from the Institute for Basic Science (IBS), South Korea, under grant IBSR028-D1.



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
