# Peer review of "Does the AO index have predictive power regarding extreme cold temperatures in Europe?"

_Natural Hazards and Earth System Sciences, 2020_

## Referee Comment (RC1) · Anonymous Referee #1 · 27 May 2020

The study is focused on modelling cold extremes by using EVT tools and covariates such as AO. The objective is to propose an approach to be used for long-term weather forecasting, although the authors tested and used just observational data. From a methodological point of view, there is nothing new except for the idea of measuring the predictive power by using the 'sharpness' criterion introduced by Gneiting in 2007. Thus, results are somehow expected. My main concerns are on the applied tests and on the readability of the manuscript. As for the former one, I suggest to use EDF-tests, while for the latter one I think the manuscript needs to be fully restructured. Results, methodological details, technical discussion, etc. are all mixed through the entire manuscript. Just to make an example, the introduction should give more emphasis on the objectives of the study, what has been already done by others, etc. rather than focusing immediately on technical details.

Some Specific Comments 20 Well, this is true if there is convergence 25 I do not see why the parameters should change from one winter to the other, especially the shape. Maybe some more words on this, would help readers to understand what the authors' view is. 60 There is no need of the first paragraph 65 It is not that evident from Fig. 1 70 Terms of the equations should be explained. 85 Again, the GEV is correct as soon as there is convergence, that is actually very difficult to test as requires very large samples 109 This is a questionable assumption if we say that external forcing and not natural variability is causing the long-term trend (or a combination of the two). Furthermore, I do not see how you can be confident with the robustness. I think it is more fair to just say there is an assumption that may be violated. 155 I suggest to used EDF-tests instead of these two as the focus is on GEV family. 160 I suggest to rephrase this paragraph. Figure 2 is not very informative Figures 7 and 8 Maybe as SOM?

---

## Author Comment (AC1) · 3 Jun 2020

We would like to thank the referee for their time and effort with reading our manuscript and for providing feedback. We quote the reviewer's points and comment on them below (starting with an indentation).

The study is focused on modeling cold extremes by using EVT tools and covariates such as AO. The objective is to propose an approach to be used for long-term weather forecasting, although the authors tested and used just observational data. From a methodological point of view, there is nothing new except for the idea of measuring the predictive power by using the 'sharpness' criterion introduced by Gneiting in 2007.

[Figure]

Thus, results are somehow expected.

In our opinion, the latter is a mistaken conclusion. Employing well-established methods does not mean that the results can be expected. If anything, the methods need to be employed and the work done because we cannot foresee the results of it. As an example, a Runge-Kutta integrator scheme is most common to integrate ODE systems. Yet, when the solution is chaotic, it cannot be anticipated without actually solving the system numerically. Or, engineering consultancy most often does not need to come up with new methods to make an analysis. Yet, firms are paying very substantial money for the analysis, simply because its results cannot be foreseen.

My main concerns are on the applied tests and on the readability of the manuscript. As for the former one, I suggest using EDF tests,

We would like to note, to start with, that in our original analysis we ended up with using the Lilliefors test, because the Matlab help file for the Kolmogorov-Smirnov test, which is an EDF-based test, commented that "The result is not accurate if CDF is estimated from the data. To test x against the normal, lognormal, extreme value, Weibull, or exponential distribution without specifying distribution parameters, use lillietest instead."

https://ch.mathworks.com/help/stats/kstest.html?searchHighlight=kstest&s_tid=doc_srchtitle#namevaluepairarguments.

We did not find a similar caution about the Anderson-Darling test (AD), another EDF-based test, and, so, following the Referee's suggestion, we have evaluated the p-values in the same fashion as in the case of the original two tests used. We used the Matlab function 'adtest':

https://ch.mathworks.com/help/stats/adtest.html

The results for the four models can be seen in the diagrams attached. It turns out that the AD is the most lenient out of the three tests pursued up to now. In particular, for

models M1, M2, M4, hardly any gridpoint sees the rejection of the GEVD form when using AD. (Only in the case of M3 do we see many gridpoints of rejection, which is similar to the results with the original two tests, not shown in the paper.) The sensible approach is a conservative one: if one of two tests rejects the GEVD, then we cannot go with that model even if the other one could not reject it. Therefore, aposteriori, the AD does not seem to be the one that we should rely on.

while for the latter one I think the manuscript needs to be fully restructured. Results, methodological details, technical discussion, etc. are all mixed through the entire manuscript. Just to make an example, the introduction should give more emphasis on the objectives of the study, what has been already done by others, etc. rather than focusing immediately on technical details.

We think that the paper has an adequate structure, which is not really out of line with the mainstream. In our opinion, it is not very constructive to try to precisely separate passages of text relating to methodology, results, literature review, etc. Methodological, technical descriptions and comments appear to us to be appropriate to make in the Introduction or the section conveying the main results.

We intended to write a short paper, and we believe that we did set out our main aims in the Introduction and also gave there a backdrop of it including references to relevant past work. We do think that the technicality of mentioning nonstationary extreme value statistics in the Introduction is necessary in order to be able to say what our work is really concerned with.

Nevertheless, we would be willing to streamline the Introduction and perhaps implement some sensible restructuring.

Some Specific Comments

20 Well, this is true if there is convergence

Convergence is taking place asymptotically. One of the authors worked together with

a senior colleague who kept speaking about "convergence setting in" or "convergence taking place". But in fact, it never does. Therefore, a GEVD model estimated from data that does not actually conform precisely to that model (even if a statistical test cannot reject it – a point made also by the referee)

http://centaur.reading.ac.uk/73386/

https://www.cambridge.org/core/books/nonlinear-and-stochastic-climate-dynamics/extreme-value-analysis-in-dynamical-systems-two-case-studies/CE02FCFC8BF95D735ECA2298D1CA8E89

will take a range of values to perform well. That is, while extreme value theory is meant to facilitate an extrapolation based on universality, such an extrapolation has its practical limits. In fact, the original pioneering work by Fisher and Tippett (1928), which we cited at this point of the text, is concerned with the problem of convergence, and therefore maybe we don't need to prompt a caveat explicitly. This topic is nowadays called Penultimate Extreme value theory (P-EVT).

25 I do not see why the parameters should change from one winter to the other, especially the shape. Maybe some more words on this would help readers to understand what the authors' view is.

This is a very useful point. Thank you. On lines 93-97 we wrote that "different Poincaré sectioning surfaces associated with different fixed values of the co-variate should yield sections of the attractor whose geometries are different, yet, their fractal dimensions (Hall and Davies, 1995) and so the shape parameter (Holland et al., 2012; Lucarini et al., 2014; Bódai, 2017) should be the same. Nevertheless, a considerable variation of finite-size estimates is possible, even if the pre-asymptotic non-GEV characteristics do not show up (Bódai, 2017; Gálfi et al., 2017)". Given that in the formula above line 25 $Z_t$ is meant to be a monthly maximum, i.e., the block size is finite, we could not possibly mean the asymptotic limit. In terms of P-EVT, the shape parameter does not need to be the same for all values of the covariate – even if it actually is in

the asymptotic limit, as we explained. Nevertheless, as we write on l 74 and 212, the shape parameter probably doesn't vary much with the co-variate, and, so, a lot of data would be needed to possibly reject the hypothesis of a constant shape parameter. We would leave $\xi(t)$ (instead of writing $\xi$ denoting a constant) in the formula in question, and add that we do not restrict generality there, and, perhaps, we could refer forward to Sec. 2.1 where we do in fact settle with the constant $\xi$ model.

60 There is no need of the first paragraph

Ok.

65 It is not that evident from Fig. 1

This is actually true. Thank you very much for pointing this problem out. We realise now that we had been primed by our preliminary figures (not shown in the paper) based on the DJF-mean AO index and corresponding mean daily minimum temperatures; one for a location in Northern Europe and one for Kiev. We attach now a diagram like that in our Fig. 1a, but with one new data line for the DJF-mean AO vs the corresponding mean daily temperature minimum. (The data points are linked for visualisation reasons only; they do not indicate chronology.) We can see that there is a correlation also for positive values of the AO index, unlike we wrote in the paper. Therefore, we propose that we replace Fig. 1a with this new figure, and modify the text on l 67-71 as follows.

"This seems to apply also to the location of Kiev, as indicated by the solid thick data line in the diagram in Fig. 1 (a). In the same diagram, a scatter plot of the daily minimum temperature in Kiev vs the monthly mean AO index (AO) provides some view also on intraseasonal variability including extremes. Our observation is that the characterisation of the AO-dependence of extremes – unlike that of the seasonal means – will have to include some nonlinearity. In particular, the overall dependence of e.g. the location parameter $\mu(AO)$ of a "nonstationary GEV distribution" is seemingly nonmonotonic."

70 Terms of the equations should be explained.

[Figure]

We are not sure which terms are not clear. Just before these equations, we are motivating the adoption of a model quadratic in the variable $AO$. That leaves all other symbols to stand for constants. We have not excused ourselves that we recycle the symbols, $\mu$, $\sigma$, $\xi$ appearing in the form of the GEVD in the Introduction (as we do not recycle), and, therefore, the meaning of these symbols should be clear. The location parameter $\mu$ was explicitly mentioned just above on l 70; and the scale parameter $\sigma$, key for our analysis, was discussed in the Introduction. Furthermore, in the revision, we would make the forward reference in the Introduction about the shape parameter $\xi$, according to the previous point made by the referee.

85 Again, the GEV is correct as soon as there is convergence, that is actually very difficult to test as requires very large samples

We agree about these two issues with convergence. That is, firstly, with finite block size the distribution of block maxima is nearly never exactly a GEVD. It is a GEVD only if the parent distribution itself is a GEVD – thanks to the max-stable property of the GEVD. Please note that this is precisely what we meant by the bracketed text on l 83. Secondly, the finite-size non-GEVD character might require a lot of data to detect. This was one of the key points of

https://www.cambridge.org/core/books/nonlinear-and-stochastic-climate-dynamics/
extreme-value-analysis-in-dynamical-systems-two-case-studies/
CE02FCFC8BF95D735ECA2298D1CA8E89

Indeed, we might know from theory that the data cannot conform to the GEVD, whether it is because of the finite block size or because of the fractality of the probability measure, yet we might not be able to detect it if there is not enough data and the irregularity is not so strong.

Not independent from an earlier point of the referee, we would revise the text here, extending our sentence on l95-97, as follows.

[Figure]

"Nevertheless, considerable variation of finite-size estimates is possible, even if the pre-asymptotic non-GEV characteristics do not show up (Bódai, 2017; Gálfi et al., 2017), which dictates a generically non-constant shape parameter, as expressed in Eq. (1) [equation number to be introduced]. Yet, we adopt a constant shape parameter model (1) [equation number will change], as it is known that the estimation of the shape parameter of a stationary process is rather sensitive to data shortage (Friederichs and Thorarinsdóttir, 2012)."

The last clause in the above is a copy from l 212-213. We are willing to take care of the details of a good restructuring and editing in the revision.

109 This is a questionable assumption if we say that external forcing and not natural variability is causing the long-term trend (or a combination of the two).

Please excuse us, we are unsure what you mean. We cannot see why the external forcing scenario (l 106-107) should jeopardize our assumption more than the decadal-scale internal variability (l 114-). Indeed, this is just an assumption, something that we have not proven. But we have two important points. One is that the assumption is rather intuitive. Just because some values of the co-variate occur more frequently in different time periods (under different "climates"), the same value might imply very similar (conditional) distributions of the observable of interest in these different periods. For example, maybe some warm Decembers will occur more frequently in the future, but even in the past the probability that the monthly maximum temperature will exceed a threshold, given the same particular value for the monthly mean, might be largely unchanged. The second point is about robustness, that you address yourself by your next point.

Furthermore, I do not see how you can be confident with the robustness. I think it is fairer to just say there is an assumption that may be violated.

We wrote on l 111-113 that if our assumption – regarding the seasonal nonstation-arity – was inaccurate, that would imply poorer predictive power in terms of a larger

scale parameter. That is, we will not see a false gain of predictive power when our assumption is inaccurate, but rather the opposite. We see it as a good conservative approach.

We should make it explicit indeed that this is generic, applying also to the long-term nonstationarity. We propose to append the paragraph on l 119 by the following sentence. "Furthermore, the conservative nature of the estimate applies generically, including the climate-change-type nonstationarity."

155 I suggest to used EDF-tests instead of these two as the focus is on GEV family.

We will make a note of the results of the Anderson-Darling test as per the above, without including figures. Again, we find that it is too lenient compared to the original two tests. Otherwise, we see no fundamental reason why EDF-tests should better suit the GEVD form. We note that the Matlab implementation of e.g. the Lilliefors test

https://www.mathworks.com/help/stats/lillietest.html

takes the Gumbel distribution as one of those that can be selected by the user.

160 I suggest to rephrase this paragraph.

We will make an attempt to do this. Perhaps it is too concise as it is now.

Figure 2 is not very informative Figures 7 and 8 Maybe as SOM?

Fig. 2a shows the large variability in winter relative to the other seasons, and that it looks as though the "outgrowth from a harmonic function is towards negative temperatures". Furthermore, both Fig. 2a and b convey information about the amount of data available. It is a key theme of our paper that data scarcity needs to be handled somehow.

We suppose that "SOM" refers to a supplementary material. Ours is a short paper and it would feel not appropriate for us to have a supplementary material with just two figures. Furthermore, we think it is an important enough point that the models do not

apply well to all locations. The referee himself/herself regarded the perceived issue with the statistical tests as one of the two of their main concerns.

**ECAD gridpoint data, lat = 50.25, lon = 30.25**

**Fig. 1.**

Interactive
comment

[Figure]

**Fig. 2.**

---

## Referee Comment (RC2) · Anonymous Referee #2 · 23 Jun 2020

I have read with interest the paper : " Does the AO index have predictive power regarding extreme cold temperatures in Europe? ". The authors claim, not surprisingly, that what they define a " native " covariate (the Temperature) is better at forecasting extreme cold temperature in winter than a dynamical co-variate, i.e. the Arctic Oscillation index. Although I appreciate the use of the non-stationary GEV model for this kind of studies, that in my opinion this paper is yet too far from being publishable in NHESS. My recommendation is to reject the paper in its current form and I encourage the authors to completely rethink their study. The two most important arguments for rejection are the following :

[Figure]

1) The results of the paper are trivial : a good representation of cold extreme temperature distribution in winter is obtained if a model which include temperature is used. Of course, another index that does not include temperature information performs poorly. How is this results useful at all for the community of seasonal forecast ? Why do the authors analyse only the AO index, instead of focusing on SST, NAO, or an index based on weather regimes computations ?

2) In the paper, the authors fail to evaluate what in the title and the abstract they announce as " seasonal predicting power " of their model. They limit their analysis to fits of distribution and tests for distributions (Lilliefors and Chi2). Standard forecast skills metrics are not applied (e.g. CRPS, RMSE). The scientific question claimed in the title, about predictive power, should be answered by : i) training the model on just a part of data (training data set), 2) testing the quality of the forecast in a testing data set (the remaining of the data set). If they really want to say something about seasonal forecasts, then the question is how to evaluate the forecast made for DJF if when the model is initialized in November or October. They should then try to run the model over ensembles of possible AO and T values and finally get a CRPS (or equivalent foreacst skills metrics).

Other major comments :

1) The abstract is totally non-informative. I would recommend to review the abstract with the following suggestions : - " extreme value statistics " of what ? Are the authors talking about weather extreme events ? Which ones ? - " large scale quantities " ? Which scales ? Large with respect to what ? I think the authors mean synoptic scales here. And what are the " quantities " ? Are the authors referring to climate indices ? - " nonlocal " with respect to what ?

2) The presentation of the data is completely displaced : The authors evaluate M1 (model 1) in lines 80-100 before presenting the data sets used (lines 140-145) and even before presenting the evaluation metrics (section 2.2). Furthermore, the authors

mix up different time scales in their analysis : daily, monthly, seasonal. It is not clear whether the data for AO are daily or Monthly ? Do you use ECAD or EOBS ? What is the time scale of the model? Daily or Monthly ?

3) The evaluation of the model M1 is just qualitative. No CRPS or other standard forecast evaluation metrics are provided, or they appear later in the text, generating great confusion.

4) The authors recognize that seasonality and non-stationarity are important and should be taken into account. They say that their methodology " does not (and probably cannot) " take into account this issue. However they did not even try to apply the standard procedure to take into account this issue : e.g. repeating the analyses by removing a linear trend on the temperature, removing the seasonal cycle and repeating the analysis. These are very elementary tests and for such a simple basic analyses they should have been implemented.

5) Another point about the evaluation metrics : Lilliefors and chi2 tests are good to assess the adherence of distributions to the targeted ones. However, they do not say anything about the dynamics and therefore the predictive power of the model in terms of forecasts. I guess the authors are well aware that reshuffling the data destroys all the dynamical features (and therefore the predictability) but preserves the distribution so the results for both tests will be identical, even with a completely random dynamics.

6) The results section consist only of few lines of figure descriptions. No attempt of explaining the geographical differences on model performance is provided. Speculative sentences with no justifications appear here and there

Minor (but important) remarks

Introduction : -L9 : you cite Tel and Gruiz but the discovery of chaos in weather forecasts dates back to Lorenz 1963 -L12 : " ignoring completely some physical quantities " which ones ? References ? -L20-25 : in the GEV definition, mu is undefined, xi is

undefined, only sigma is defined. These equations should be numbered.

Methodology :

LL80-100 : as said before, this evaluation is completely subjective and no quantitative analysis are provided. Note also that Yiou et al. 200 GRL, Ferranti et al. 2015 QJRMS, and Faranda et al. 2015 (Clim Dyn) have suggested that there is a non-trivial relationship between the AO patterns and time series and the predictability of the weather. They suggest that a simple time-series analysis should be discarded in favor of more comprehensive dynamical systems approaches.

LL105 " Seasonality: the different months of the winter should have different climatologies (Bódai and Tél, 2012). " I do not think this (auto)citation is pertinent here.

LL114-115 : " Furthermore, beside the climate-change-type nonstationarity, playing out on multidecadal time scales, there should be considerable internal variability on multidecadal time scales, too. ". this statement is too qualitative, how do the authors support it ?

Results :

LL171-176 Would the authors expect anything different ?

Discussion :

LL 220-222 I would be more careful and respectful citing the work of other colleagues as " ironic "

LL223-226 The authors mix up again (partially admitting) their confusion about predicting power, predictability, seasonal forecasts. . . I want to stress once again that distributions (and all the metrics associated) are totally insensitive too reshuffling. Metrics for predicting power and/or predictability are based on time evolution.
* * *
2020-117, 2020.

---

## Author Comment (AC2) · 22 Jul 2020

We thank the Reviewer for taking their time to read our paper and providing feedback on it. It is regrettable that they recommend rejection, but also that they either didn't have the chance or weren't willing to engage in a discussion on the points of their criticism, because we think that their major concerns are misplaced, which could perhaps be clarified at an earlier stage. In fact, the proportion and nature of the misplaced criticisms is such that we find it hard to possibly account for it by randomness. In any case, we address the individual points below in indented paragraphs.

Otherwise, we acknowledge that the reviewer comments prompted us to think over

[Figure]

certain issues carefully, which lead us to propose some additional analysis as we spell it out under point 4) of the "Other major comments".

I have read with interest the paper: "Does the AO index have predictive power regarding extreme cold temperatures in Europe?". The authors claim, not surprisingly, that what they define a "native" covariate (the Temperature) is better at forecasting extreme cold temperature in winter than a dynamical co-variate, i.e. the Arctic Oscillation index. Although I appreciate the use of the non-stationary GEV model for this kind of studies, that in my opinion this paper is yet too far from being publishable in NHESS. My recommendation is to reject the paper in its current form and I encourage the authors to completely rethink their study. The two most important arguments for rejection are the following:

1) The results of the paper are trivial: a good representation of cold extreme temperature distribution in winter is obtained if a model which include temperature is used. Of course, another index that does not include temperature information performs poorly. How is this results useful at all for the community of seasonal forecast? Why do the authors analyse only the AO index, instead of focusing on SST, NAO, or an index based on weather regimes computations?

This is essentially the same concern that also Reviewer #1 had. We cannot deny that we foresaw that the local temperature ($T$) co-variate would have more "predictive power" (in the sense as explained in the paper) than the nonlocal AO index (AOI). In actual fact, the second author suggested the use of the AOI, and the first author was more interested in the connection between the means and extremes of the same quantity (thinking about connecting large deviations theory and extreme value theory). What is not trivial is the actual numerical figures of the superiority of $T$, or that the AOI and $T$ together do not have better predictive power than $T$ alone (in the absence of enough data).

Of course, there are many nontrivial but useless or uninteresting questions to consider. Accordingly, it is a separate question whether it is worth determining the predictive powers of both the AOI and $T$ from the point of view of seasonal forecasting in particular. As we have it in the Discussions section, our study relates in an interesting way to the results of Dobrynin et al. (2018). They found that the seasonal forecast of monthly mean surface air temperature in Europe can be considerably improved if the forecast ensemble is sampled by retaining ensemble members according to whether the NAO is well-predicted by a member. (We would think that the AOI would perform similarly to the NAO. However, perhaps we should redo our analysis, switching from the AOI to the NAO, in order to have a good and so more informative fit with Dobrynin et al. (2018).) That is, despite that the AOI has a modest predictive power in the sense of our paper, and based only on statistical relationships established from data, using a circulation model as part of a forecast system, extremes might be better predictable "concerning the AOI" (in the sense of a NAO- or AOI-based ensemble sampling) IF the mean temperatures have a sufficiently better predictive power (again in the sense of our paper). Whether it is sufficient we have not determined, as we haven't evaluated a forecast system, but our paper indicates that there is a potential for it given the finding of the very considerable superiority of $T$ as a precursor over the AOI.

These two points of ours do in our opinion negate the first of the main concerns of the Reviewer.

Regarding the further suggestion of the Reviewer of using some quantity characterising the SST of the North Atlantic as a co-variate, we do think that it is meaningful, and that it might even yield a better data-based extreme cold prediction skill than using the AOI as a covariate, or, better than the NAO-based subsampling of Dobrynin et al. (2018) combined with the use of $T$ as a co-variate. However, we think that it lends a good focus to the paper if we consider only two of the co-variates and make a close connection with the paper by Dobrynin et al. (2018). Concerning "weather regimes", we doubt that such a crude concept, a discrete variable that can take only a few values, as opposed to a full range of the real numbers, can do better. We cited (van den

Besselaar et al. 2009) as an example of such a study (which is actually also a rather short paper).

Regarding the Reviewer's doubt that our result concerning the "predictive power" of the co-variates only (not the skill of a forecast system) can be of interest to the seasonal forecast research and practice community, we don't see why not, given that one of the authors of (Dobrynin et al. 2018) had read our manuscript and expressed interest in collaboration. We do understand, however, that a competition and controversies exist between forecast system developer groups.

Finally, we think that the Reviewer's statement "a good representation of cold extreme temperature distribution in winter is obtained if a model which include temperature is used" should be backed up. It is commonly said that numerical weather prediction (NWP) models that are calibrated to represent TYPICAL events well do NOT represent extreme events well. This problem is eliminated when we use observational data and forecast only typical values of a large-scale quantity like the AOI. We do realise that concerning a single month, e.g. December, we have only 70 monthly maxima data points from the ECAD (from a 70-year period), which is a comparable number to the 51-strong ensemble of the ECMWF seasonal forecast system. However, observational data is not suffering from biases by definition, of which two sources should be considered: 1. unrealistic model physics; 2. finite spatial resolution of the NWP model. Given a particular site where perhaps a long temperature record is kept, because of the importance of a facility or the curiosity of the resident, a "site-targeted" forecast can be generated just by using the forecast of large-scale nonlocal quantities. Regarding points 1. and 2., see L37-40. Considering this, instead of the AOI, the co-variate of the monthly-mean temperature in a GRIDPOINT of the NWP model nearest to our site should not be considered precisely "native" or local.

2) In the paper, the authors fail to evaluate what in the title and the abstract they announce as "seasonal predicting power" of their model. They limit their analysis to fits of distribution and tests for distributions (Lilliefors and Chi2). Standard forecast skills

metrics are not applied (e.g. CRPS, RMSE). The scientific question claimed in the title, about predictive power, should be answered by : i) training the model on just a part of data (training data set), 2) testing the quality of the forecast in a testing data set (the remaining of the data set). If they really want to say something about seasonal forecasts, then the question is how to evaluate the forecast made for DJF if when the model is initialized in November or October. They should then try to run the model over ensembles of possible AO and T values and finally get a CRPS (or equivalent foreacst skills metrics).

The Reviewer has a misunderstanding here. This pertains also to our answer to point 1) above. We went to tedious length in our paper explaining what we mean by "predictive power" of a co-variate, or, more precisely, a nonstationary GEV distribution model featuring one or more co-variates (see e.g. L25-34 including footnote 1). A "forecast skill score" like CRPS is something else, and it pertains to an actual method of prediction – and it is evaluated or estimated by MAKING predictions. We do not make predictions of the temperature extremes given that we do not use a forecast system to forecast AOI or the monthly mean $T$ (see L30). We had always envisaged carrying on with our work in the direction as the Reviewer suggested (see L226). However, we cannot agree with him/her that our first analysis of the predictive powers alone is unsubstantial and uninteresting. In our view, this is a point very much worth separating, also in the sense of publishing it separately.

Please note the Reviewer is misquoting the title of our paper; we did not use the word 'seasonal'.

Other major comments :

1) The abstract is totally non-informative. I would recommend to review the abstract with the following suggestions:

We cannot agree with the Reviewer that the Abstract is noninformative, however, we could improve it by adding quantitative information. There are two statements in the

Abstract:

1. $T$ has a much larger predictive power than AOI. Here we can ADD, as per Fig. 6a, that in terms of our measure of predictive power (the difference between the "stationary and nonstationary scale parameters", L28, 164), $T$ has at least 3 times larger predictive power, but it is 10 times larger in more than half (we can get the actual figure) of Europe (as covered by the ECAD data).

2. Using AOI and $T$ together does not yield a larger predictive power than using $T$ alone, based on a 70-year-long data record.

The following list of concerns of the Reviewer is not difficult to respond to. In fact, we find it puzzling how it is possible that the same simple points did not occur to the Reviewer himself/herself.

- "extreme value statistics" of what?

The emphasis of the first sentence is that we use NONSTATIONARY extreme value statistics (EVS). Such a methodology can have other forecast applications, say, the forecast of warm temperature extremes in the USA with e.g. the Nino3 index as a covariate. In fact, we are proposing that teleconnections with respect to extremes – as opposed to averages, as usually considered – can be framed by nonstationary EVS. The next sentence is informing the reader WHAT extremes we are concerned with in particular in the paper.

Are the authors talking about weather extreme events?

The first sentence informed the reader that our work is motivated by SEASONAL forecasting. It is hard to conceive that anything else but extreme WEATHER events can be implied by this.

Which ones? - "large scale quantities"? Which scales? Large with respect to what? I think the authors mean synoptic scales here. And what are the "quantities"? Are the

authors referring to climate indices?

We do not understand why the Reviewer is asking this. It is clearly written (see statement 1. as we restated above) that it is the "monthly mean daily minimum temperature" and "monthly mean AO index" whose predictive power we are evaluating.

- "nonlocal" with respect to what?

We really don't understand why the Reviewer thinks that this question needs asking. The Reviewer does not say explicitly that he/she did really not understand what we meant, and we assume that he/she did, and we do not think that anyone of the targeted audience of this paper would not know what we mean such that more words would be required. The AOI is the principal component of an EOF. Obviously, it does not pertain to a particular location like an extreme cold temperature somewhere in Europe that we want to forecast, therefore, we call it "nonlocal".

We do not think that reasonably brief exposition should be a privilege of senior scientists.

2) The presentation of the data is completely displaced: The authors evaluate M1 (model 1) in lines 80-100 before presenting the data sets used (lines 140-145) and even before presenting the evaluation metrics (section 2.2).

We do not think that the fact that the Reviewer would pursue their exposition of this topic in a different way (according to his/her current understanding) is sufficient basis for criticism, even to the length of recommending the rejection of the paper. For the first point here, we note that what particular data set we used is mentioned briefly in the Introduction, and also in the caption of Figure 1. There is really no obstacle to understanding the exposition of our methodology, pertaining to a particular location, as it appears before line 140, from which point on we give more detail about the data set. In fact, these new details concern the spatial resolution, which a reader does not need to know about before proceeding to mapping out things extensively, in the whole

spatial range of the data set.

We suspect that there is a misunderstanding about what is in Sec. 2.2. We suspect that the Reviewer refers to the "predictive power" of a model like M1 by the expression "evaluation metrics", because he/she uses that expression just after writing "evaluate M1". However, Sec. 2.2 is not evaluating model performance but rather reliability, i.e., if there is a reason to think that the inferred/estimated scale parameter is biased. It is what is usually referred to as "model error", something that could lead to deteriorating prediction skill. However, we do not evaluate the prediction skill of a forecast system and, so, we have to (and did) at least indicate, in an additional analysis step, if our figures of the predictive power of a model are reliable.

Furthermore, the authors mix up different time scales in their analysis: daily, monthly, seasonal.

We do not understand what the Reviewer means.

It is not clear whether the data for AO are daily or Monthly?

It is already written in the Abstract that it is the monthly-mean AOI. Furthermore, see L45, 68. We can locate these mentions, with minimal effort, by searching the pdf file for the word 'monthly'. Also, Fig. 1 is the main visual tool to explain our "gridpoint-wise" methodology. The axis label also clearly says 'DJF monthly mean AO index'. What 'DJF' refers to must be clear by looking at the legend explaining the meaning of the markers in color.

Do you use ECAD or EOBS?

We quote the following from https://www.ecad.eu/download/ensembles/download. php: "the ÂăENSEMBLESÂădaily gridded observational dataset for precipitation, temperature and sea level pressure in Europe called E-OBS" and "the ECA&D staff will maintain and update the E-OBS gridded dataset". From this it appears to us that the data that we used could be referred to either as ECAD or E-OBS data, both name

pertaining to a project title. Perhaps ECAD is more generic in that it also refers to the instrumental-, or also called station, data record; and E-OBS refers only to the derivative interpolated gridded data.

It is true that some refining of our reference to data used is in order, thank you for signaling this to us. On L36 we will delete "E-OBS". On L41-42 we will write "...drawing on the E-OBS gridded data set of the European Climate Assessment & Dataset (ECAD, www.ecad.eu) project" . In the figure caption of Fig. 1 we change "ECAD" to "E-OBS"; and in the caption of Fig. 3 we change "ECAD data" to "E-OBS gridded data".

What is the time scale of the model? Daily or Monthly?

We do not understand what model is meant. Perhaps the above clarification renders this question redundant too.

3) The evaluation of the model M1 is just qualitative.

The word 'qualitative' in the above does not carry a meaning to us.

No CRPS or other standard forecast evaluation metrics are provided, or they appear later in the text, generating great confusion.

No, they do not appear later, which is clearly stated in the Introduction (L34). See also our response above to the first of the "main concerns" (point 1)). We think that the confusion is not on our side.

4) The authors recognize that seasonality and non-stationarity are important and should be taken into account. They say that their methodology "does not (and probably cannot)" take into account this issue. However they did not even try to apply the standard procedure to take into account this issue: e.g. repeating the analyses by removing a linear trend on the temperature, removing the seasonal cycle and repeating the analysis. These are very elementary tests and for such a simple basic analyses they should have been implemented.

[Figure]

This is the first point of concern that goes beyond style, clarity and relevance, questioning the reliability of our actual results. We mention in passing, with reference to the first main point of concern 1) on triviality, that such a concern about reliability pertains to the quantitative aspect of the results, which we think constitutes an acknowledgement of the nontriviality of theirs, thereby contradicting the said point 1).

There are in fact two issues here. 1. One is a question about how we can deal with nonstationarity. 2. The other one is a question asking that if we ignore nonstationarity, like we did, then what effects it has on our results as for the predictive power. Regarding 1., it is important to appreciate that to be able to come up with a model like e.g. M1, we need observational data, which is in the past. In the case of nonstationarity too, we cannot consider any other possibility but relying on past observations, from which we somehow glean a pattern about changes to come, at least in the near future when we are intending to make use of the model. The pattern gleaned can certainly be only an approximation considering that the true nonstationarity, i.e., forced response, can only be determined from an (observational) ensemble (Drotos et al. 2015), which is not available even in the past. This is why we wrote in the paper that our method cannot deal with nonstationarities. It takes some luck that the true form of nonstationarity, with respect to some statistics of some observable quantity, approximates a simple form like a linear or a quadratic trend. Such a long-term nonstationarity can be represented in the model with time now as an explicit co-variate, associated with two extra parameters in the case of a linear trend of one GEV parameter. Then, a parameter inference can be performed by Maximum Likelihood Estimation (MLE), and subsequently we can perform a statistical test, e.g. the Lilliefors test (having transformed the random variables associated with each data point to a "standard" Gumbel rv., as described in Sec. 2.2) to see if the assumed form of nonstationarity is a reasonable approximation. Furthermore, one would also take a look at the significance of the parameter representing the slope of the trend. This would be based on the confidence intervals of the MLE. If it is not significantly nonzero, then there is no point in assuming nonstationarity.

In fact, we PROPOSE that as part of the revision of our paper, if we are given the chance, we will carry out the analysis that is outlined just above. That is, we prefer not to remove a linear trend upfront, as the Reviewer and others had suggested to us. We emphasize that there is always a trend in a finite time window due to internal variability, and, when the climate is actually stationary, it is a mistake to remove such a trend. This point was made e.g. in (Drotos et al. 2015). In addition to the long-term nonstationarity, we can check the importance of the seasonal nonstationarity too in a similar manner. We could assume a quadratic dependence of a parameter on time, which could achieve a nonmonotonic time-dependence (certainly the most plausible in the case of the location parameter). This would entail the inference of 3 more parameters per GEV distribution parameter. However, we have 3 months (DJF) to distinguish, and therefore a quadratic model will not be more parsimonious than just assuming that the GEV distribution parameters are different between the months. This means that instead of pooling data from 3 months, we just fit a model to December-only, January-only, February-only data, respectively. That is, we fit the same model not to 3*70=210 data points, but to 70 data points at a time. This should increase the confidence interval of the parameters. In any case, we would need to check if the D J F parameters (all of $\mu, \sigma, \xi$ in the most generic analysis) are statistically significant. If not, again, there is no reason not to pool the DJF data (like we did).

However, if the result indicates a significant nonstationarity, either seasonal or long-term, it might still be misleading. It might be so that it is not due to actual nonstationarity, i.e., the explicit time-dependence of the dynamical system, but rather long-term internal variability. For example, the time-dependence of parameters, as shown by Fig. 1 of

https://agupubs.onlinelibrary.wiley.com/doi/full/10.1029/2019GL085881

in the ENSO model given by eq. (1) is not due to true nonstationarity but internal variability. This is why we wrote in our paper that our method probably cannot deal with nonstationarity. This means that even if the climate is stationary, having "trained" a model, i.e., inferred its CONSTANT parameters, on some historical data, it might not

remain accurate if the climate state transitions to a new regime as part of its internal variability. An example of this is the declining correlation, as measured by moving-window correlation coefficients, between some ENSO index and the "All-India Summer Monsoon Rain" since about 1980.

https://doi.org/10.1126/science.284.5423.2156

However, observational data sets were found by Yun & Timmermann (2018)

https://doi.org/10.1002/2017GL076912

to be consistent with a stationary process. The true forced response of the ENSO might even go the other way:

https://doi.org/10.1175/JCLI-D-19-0341.s1

Nevertheless, even if nonstationarity cannot be detected, as a result of data scarcity, we are rather sure that there is true seasonal nonstationarity. In fact, because of this, working with monthly maxima is already a "practical compromise". The correct thing would be in fact is picking the maximal element of an evolving ENSEMBLE, instead of picking the maximal element in a time window in which the ensemble does evolve ("transform"). This is the methodology that we adopt in analysing the forced response (seasonal and long-term) of EVS in an ongoing study concerning an Earth System Model, working with daily maxima. We can confirm that any of the GEV distribution parameters can change even within a month. An important question is if this is a nonasymptotic finite block size behaviour. In this regard we need to make some corrections to our response to Reviewer #1 and revise the paper around L95 on the constancy of the fractal dimension and, so, shape parameter. That applies only to autonomous systems. As for nonautonomous systems, Ledrappier and Young (1988)

https://link.springer.com/article/10.1007%2FBF01218383

showed that, curiously, the fractal dimension remains unchanged if the forcing is sufficiently weak so that the topology of the phase space at least is stationary, despite

that the ensemble itself is evolving. (Although finite-size dimension estimates have a nonnormal fluctuations (Bodai et al. 2011, DOI: 10.1103/PhysRevE.83.046201).) However, under generic forcing, the topology of the snapshot attractor does not need to stay unchanged, some bifurcations can occur (which is a mathematically unexplored area), and therefore the fractal dimension and so the shape parameter of the GEV distribution will also not stay constant. We strongly believe that the seasonal forcing of the Earth climate is so strong that the fractal dimension changes, and therefore so does the true asymptotic value of the shape parameter.

Regarding 2., we made the claim in the paper on L111-113 that the predictive power would be CONSERVATIVELY estimated, approximating the correct value of the scale parameter from above (with a larger value corresponding to less predictive power). This is rather very intuitive, we think. It is easy to see that the sample variance of draws taken from random variables of equal variance but unequal means is expected to be strictly larger than the true variance. The situation with long-term and seasonal nonstationarity of the GEV distribution is very similar. Nevertheless, we PROPOSE to demonstrate numerically that ignoring nonstationarity inflates the scale parameter. We pick a model like e.g. M1, with fixed parameters of the AO-dependence, and introduce seasonal and long-term nonstationarity with time being an explicit co-variate. Then, we sample this process over 70 years, and create many realisations of such a time series. For each realisation we estimate the nonstationary scale parameter $\sigma$ (predictive power) fitting a model that does NOT feature time explicitly as a co-variate (ignoring nonstationarity). This will produce a distribution of the scale parameter, which, with enough number of realisations, will enable us to reject the null-hypothesis, that $\sigma$ can be as small as the true value that we assigned to the model, at an arbitrary significance level.

5) Another point about the evaluation metrics: Lilliefors and chi2 tests are good to assess the adherence of distributions to the targeted ones. However, they do not say anything about the dynamics and therefore the predictive power of the model in terms

of forecasts. I guess the authors are well aware that reshuffling the data destroys all the dynamical features (and therefore the predictability) but preserves the distribution so the results for both tests will be identical, even with a completely random dynamics.

Unfortunately, we do not understand exactly what the Reviewer means here. We took great care in our paper to not use expressions like "prediction skill" or "predictability" when we mean the "predictive power" of a covariate, or, more precisely, that of a nonstationary GEV distribution featuring some particular co-variate-dependence structure like e.g. M1. Only (complete) forecast systems (able to make a forecast) have a "prediction skill" quantified e.g. by CRPS. We outlined on L28-33 of the Introduction that in our envisaged application the forecast system would be used to forecast the co-variate – not the extreme value statistics directly. We do not perform the forecast of the co-variate and therefore we cannot be concerned with evaluating a "forecast skill".

We are not sure in particular what data is meant when speaking about "reshuffling". It is important for the parameter inference concerning a model of nonstationarity like M1 that CORRESPONDING (with respect to the year that they both belong to) co-variate and monthly minimum temperature data points are passed to the MLE algorithm. If the correspondence was destroyed, by e.g. shuffling only the co-variate time series, then the inferred parameter values would be such that they would mean no co-variate-dependence (for example, $\mu_{1,AO} = 0$ in eq. (2)). That is, we would recover what we called (on top of page 2 – there was an issue with the line numbering here) the "climatological GEV distribution". This can be easily seen by looking at Fig. 1, imagining that the data points are randomly displaced along the horizontal axis. This would yield a kind of horizontal homogeneity. (Therefore, a co-variate that is uncorrelated with e.g. the AOI of SOME predictive power, would have NO predictive power, because the associated nonstationary GEV distribution would be identical to the stationary/climatological one.) We wonder perhaps that the Reviewer didn't realise that shuffling would not yield the same M1 as what we have without shuffling. (The distribution is NOT "preserved" upon shuffling.)

[Figure]

6) The results section consist only of few lines of figure descriptions. No attempt of explaining the geographical differences on model performance is provided. Speculative sentences with no justifications appear here and there

There is one item of "speculation" that occurs to us in the Results section. Namely, that the GEV distribution is perhaps rejected when the estimated shape parameter $\xi$ is positive. However, this should be viewed as a hypothesis, which we actually do not leave untested. We show that even if this hypothesis is sensible, it is not correct. This actually allows us to conclude that the GEV distribution is not rejected because the block size is too short and the distribution of block maxima is not yet approximating the GEV form (which should have a negative shape parameter based on theory), but because either 1. the model, e.g. M1, is not a good enough approximation or 2. there is nonnegligible nonstationarity within DJF or in the 70 years of the data span. Otherwise, we think it is an interesting fact that GEV distributions with positive shape parameter are not rejected despite that we know from theory that this cannot be the asymptotic value.

We do not think that it is absolutely necessary to find some sort of "geographical explanation" for the presented geographic distributions of the estimated quantities. One might not find a satisfactory answer for an excessively long time, which should not hold back the publication of the rest of the results if they have otherwise scientific merit.

Minor (but important) remarks

Introduction :

-L9 : you cite Tel and Gruiz but the discovery of chaos in weather forecasts dates back to Lorenz 1963

If the Reviewer cares to look into the cited book, they will find that the book gives a lot of credit to Lorenz. We did mean to cite a textbook on chaos here, not necessarily focused on chaos in the weather.

-L12: "ignoring completely some physical quantities" which ones? References?

Thank you for pointing to this shortcoming. As far as we can retrace our thoughts when writing this, we probably thought of wind velocities, exemplified by e.g. (Savli et al. 2019)

https://rmets.onlinelibrary.wiley.com/doi/full/10.1002/qj.3634

However, we do realise that wind velocities as a physical quantity are clearly measured (somewhere), and, so, this example too pertains to the sparsity of observations. Given that we are not aware of actual instances, while it might be the case, we will remove the statement about not measuring certain physical quantities.

Otherwise, perhaps what comes closest to what we wanted to say is to do with "observation operators"

https://www.ecmwf.int/en/newsletter/162/meteorology/progress-towards-assimilating-cloud-radar-and-lidar-observations

The problem of sparsity could be mitigated by measuring some quantities other than those that are represented by prognostic variables, as long as those quantities can be diagnosed via some extension of the NWP model. A further example would be:

Buontempo, Carlo & Jupp, Adrian & Rennie, Michael. (2008). Operational NWP assimilation of GPS radio occultation data. Atmospheric Science Letters. 9. 129 - 133. 10.1002/asl.173.

-L20-25: in the GEV definition, mu is undefined, xi is undefined, only sigma is defined. These equations should be numbered.

This is not a mistake that we made but a choice. One does not need to write down everything that they know. The parameters $\mu$, $\sigma$, $\xi$ appear in a functional form and that is enough at this point. The name-calling of e.g. "location parameter" does not give real substance. An interested and mathematically trained reader will notice anyway that $\mu$ can shift the distribution along the z-axis. This is a similar situation to how a line

or a point does not need to be defined, but they can be the (arbitrary) interpretations of Euclid's axioms.

Methodology:

LL80-100: as said before, this evaluation is completely subjective and no quantitative analysis are provided.

   We do not understand what the Reviewer means by "subjective" regarding L80-100. Neither do we understand why the labeling "no quantitative" is valid criticism. For example, someone can employ the rule of logical implication: "If A then B" implies "If not B then not A". It is not quantitative as such, but it's nevertheless part of the scientific method. A sound argument is usually not quantitative but "just" applies the rules of logic to reach a conclusion from a set of premises. Arguments are the most basic elements of scientific exposition.

Note also that Yiou et al. 200 GRL, Ferranti et al. 2015 QJRMS, and Faranda et al. 2015 (Clim Dyn) have suggested that there is a non-trivial relationship between the AO patterns and time series and the predictability of the weather. They suggest that a simple time-series analysis should be discarded in favor of more comprehensive dynamical systems approaches.

   We think that the Reviewer does not concern very deeply the relevance of the quoted three papers to the problem that our manuscript addresses. We have no doubt that using more than one variable, instead of only the AO index, perhaps something to do with the loading pattern of the AO, would improve the predictive power or the predictability. However, would this be the case if only 70 years of data is available? We recall that one of the main conclusions of our paper is that more sophistication (using both the AO index and local mean temperature $T$) might not be better than something simpler, i.e., using just one variable ($T$). If the Reviewer thinks that there is a more sophisticated approach that is better than ours for the prediction of extreme cold winter temperatures in Europe in particular, then the onus is on him/her to do that work and convince others

(and eventually get it published).

A casual reference to papers like this, and the use of an (familiar but) empty expression like "comprehensive", cannot be accepted as a proper criticism. Also, the word "discard" seems to us very much suggestive of rejecting our paper, which would be in our opinion not "fair play" (given also the anonymity of the Reviewer).

LL105 "Seasonality: the different months of the winter should have different climatologies (Bódai and Tél, 2012)." I do not think this (auto)citation is pertinent here.

It is unfortunate that the Reviewer does not explain why', and does not suggest another reference. Of course, it is a very intuitive matter that different seasons have different climatologies. However, we cited this paper because 1. it is the first paper that makes the claim that the snapshot/pullback attractor is the mathematical object that expresses the forced response of the climate system, and 2. the paper is also concerned with the seasonality of extremes in particular (even if in a toy model). We can also ask: why is the citation of (Bódai and Tél, 2012) regarding seasonality more inappropriate than the (self)citation of (Drotos et al., 2015) regarding long-term climate change?

LL114-115: "Furthermore, beside the climate-change-type nonstationarity, playing out on multidecadal time scales, there should be considerable internal variability on multidecadal time scales, too.". this statement is too qualitative, how do the authors support it?

Thank you for pointing this out. Perhaps we should change "should be" to "could be". We cannot quantify this issue based on the ECAD data. Under point 4) above we mentioned that a "significantly" nonzero estimate of a model parameter implying explicit time-dependence might not mean actual nonstationarity. (We use quotation marks in "significantly" because it could be a false result; in terms of a statistical test, the assumptions of the test, like serially uncorrelated residuals, might not be actually all satisfied, which might not be possible to actually detect from our finite data. Unfortunately, this kind of issue is all too often overlooked; p-values of trends are often given without the assurance that the assumptions of the test were checked.) The effect of long-term internal variability can only be quantified using an ensemble in the case of a truly nonstationary/nonautonomous system, or a very long data record in the case of a stationary climate.

In the case of winter cold extremes in Europe, the Atlantic Multidecadal Oscillation

https://climatedataguide.ucar.edu/climate-data/atlantic-multi-decadal-oscillation-amo

could perhaps give rise to a "significant" explicit time-dependence of the models (e.g. M1, or even M2).

Results:

LL171-176 Would the authors expect anything different?

It is not completely clear to us if the Reviewer is asking the question form a qualitative or a quantitative point of view, but we think it is the former, and in that regard we note that we have already acknowledged the situation. On the other hand, as we have also said, there is no way that one could foretell the quantitative result. As we proposed above, it is this very figure, Fig. 6(a), that can contribute some quantitative description of our finding to the Abstract.

Discussion :

LL 220-222 I would be more careful and respectful citing the work of other colleagues as "ironic"

There is a misunderstanding by the Reviewer here. There is nothing disrespectful said about the work of Dobrynin et al. (2018), even if a casual look would suggest that. In fact, we hail the result of Dobrynin et al. (2018) because it means that even if the NAO does not have such a good predictive power in terms of our analysis (statement pending on the check that NAO is similar to AOI in this respect), a "NAO-based subsampling" can lead to an improved prediction of the mean temperature, which does have a much better predictive power concerning European winter cold temperature extremes. What is "ironic" is that it is the NAO – rather than some other quantity or method – that enables the better predictability of the mean temperature.

To the contrary of the Reviewer's suggestion, one of the authors of (Dobrynin et al. 2018) had read our manuscript and responded (in writing) very positively to our very remark that their finding is surprising and ironic.

LL223-226 The authors mix up again (partially admitting) their confusion about predicting power, predictability, seasonal forecasts: : : I want to stress once again that distributions (and all the metrics associated) are totally insensitive too reshuffling. Metrics for predicting power and/or predictability are based on time evolution.

For one thing, according to our response further above, we think that the Reviewer fails to recognise that the shuffling of the co-variate time series would yield the climatological distribution, meaning no predictive power, which btw. completely satisfies intuition.

It appears to us that the confusion is on the Reviewer's side. Rather than "mixing things up", we differentiate carefully between our concept of "predictive power" and the "skill of a probabilistic forecast". It is the following part of the manuscript (L222-226) that could give the impression of confusion:

"However, it is probably the circulation model to be credited for, bridging the gap between the two co-variates considered. Perhaps the concept of the "predictive power" of a quantity can be defined less restrictively than how we did it for the present study, allowing possibly for mixing it with the concept of "predictability". This way the answer to the question posed by the title of our paper might be rather different."

However, it is important to appreciate that this appears in a discussion section. We proposed a definition of the "predictive power" in the Introduction, which was wellmotivated, and distinguished precisely from the concept of "forecast skill". In the discussion we suggest a possible way to change or extend the concept of predictive power, WITHOUT actually advocating it, and suggest in CONDITIONAL terms that then the predictive power of the AOI – answering the question in the paper's title – could be rather different. It is most common – seemingly even desired – in papers to answer firmly some questions and suggest some other questions that could have the potential to foster further progress. There are papers with questions even in the title. If there was a definite answer to those questions, then we think that rather the answer to that should provide the title.

We give some more details now about our suggestion of changing or extending the definition or concept of predictive power. Our original concept of the predictive power of the AOI is based on the conditional probability density $p(E|A)$, where $A$ denotes the AOI and $E$ denotes extremes (there is no need to be more accurate here for the argument). Let us consider the law of total probability: $p(E|A) = \int dT p(E|T) p(T|A)$. In the latter $T$ is the other co-variate, alternative to $A$. $p(T|A) = p(T,A)/p(A)$ is a climatological structure and can, therefore, also be established from a long data record. The new or extended definition of the predictive power can instead be based on the conditional probability density $p(E|A_*, S) = \int dT p(E|T) p(T|A_*, S)$, which is associated with a (hypothetical) forecast ensemble $p(T|A, S)$ (of infinite size, $N$) where $S$ denotes the state of the system at the launch of the forecast simulation (say, with 2 months lead time), and $A_*$ denotes the true value of $A$. That is, $p(T|A_*, S)$ denotes a slice of the forecast ensemble – which clearly has in general a marginal distribution $p(A|S)$ of nonzero width – at the true value $A_*$. The predictive power can then be defined, as before, as the difference between the scale parameters associated with $p(E|A_*, S)$ and the climatological one, respectively, but averaged subsequently wrt. $S$. For any $A_*$ (and $S$), the width of the distribution $p(T|A_*, S)$ should be smaller than that of the climatological one $p(T|A = A_*)$, which implies a greater predictive power compared to the original. However, this might not all be a predictive power of $A$, because even the marginal distribution $p(T|S)$ of the forecast distribution $p(T|A, S)$ might have a smaller

width than the climatological one. This could be simply due to a "predictability" of $T$ (i.e., the forecast ensemble has not fully expanded wrt. $T$). Therefore, in order to isolate the predictive power of $A$ (in the new setting of employing a forecast system) could be done by taking the difference of the scale parameters associated with $p(E|A_*, S)$ and $\int dT p(E|T) p(T|S)$, respectively (before averaging wrt. $S$). As a check, with infinite lead time, we recover our original definition. However, it is not clear to us whether it does actually make a difference if the lead time is finite.

On reconsideration of our assessment of the finding of Dobrynin et al. (2018) as "surprising (and ironic)", based on the above, we note that the fact that we find a predictive power of the AOI wrt. extreme cold temperatures in Europe should perhaps not leave us too surprised that it would have a predictive power (in terms of the new/extended definition) also 1. wrt. mean temperatures and 2. involving a NWP model. Still, we envisage that this fact can be exploited in our proposed seasonal forecast scheme of extremes via a co-variate where the co-variate-dependence is established from observational data (i.e., the extremes are not directly forecast by the NWP model).